# Few-Shot Learning with Graph Neural Networks

**Victor Garcia**[*]
Amsterdam Machine Learning Lab
University of Amsterdam
Amsterdam, 1098 XH, NL
`v.garciasatorras@uva.nl`

**Joan Bruna**
Courant Institute of Mathematical Sciences
New York University
New York City, NY, 10010, USA
`bruna@cims.nyu.edu`

## Abstract

We propose to study the problem of few-shot learning with the prism of inference on a partially observed graphical model, constructed from a collection of input images whose label can be either observed or not. By assimilating generic message-passing inference algorithms with their neural-network counterparts, we define a graph neural network architecture that generalizes several of the recently proposed few-shot learning models. Besides providing improved numerical performance, our framework is easily extended to variants of few-shot learning, such as semi-supervised or active learning, demonstrating the ability of graph-based models to operate well on 'relational' tasks.

## 1 Introduction

Supervised end-to-end learning has been extremely successful in computer vision, speech, or machine translation tasks, thanks to improvements in optimization technology, larger datasets and streamlined designs of deep convolutional or recurrent architectures. Despite these successes, this learning setup does not cover many aspects where learning is nonetheless possible and desirable.

One such instance is the ability to learn from few examples, in the so-called few-shot learning tasks. Rather than relying on regularization to compensate for the lack of data, researchers have explored ways to leverage a distribution of similar tasks, inspired by human learning Lake et al. (2015). This defines a new supervised learning setup (also called 'meta-learning') in which the input-output pairs are no longer given by iid samples of images and their associated labels, but by iid samples of collections of images and their associated label similarity.

A recent and highly-successful research program has exploited this meta-learning paradigm on the few-shot image classification task Lake et al. (2015); Koch et al. (2015); Vinyals et al. (2016); Mishra et al. (2017); Snell et al. (2017). In essence, these works learn a contextual, task-specific similarity measure, that first embeds input images using a CNN, and then learns how to combine the embedded images in the collection to propagate the label information towards the target image.

In particular, Vinyals et al. (2016) cast the few-shot learning problem as a supervised classification task mapping a support set of images into the desired label, and developed an end-to-end architecture accepting those support sets as input via attention mechanisms. In this work, we build upon this line of work, and argue that this task is naturally expressed as a supervised interpolation problem on a graph, where nodes are associated with the images in the collection, and edges are given by a trainable similarity kernels. Leveraging recent progress on representation learning for graph-structured data Bronstein et al. (2017); Gilmer et al. (2017), we thus propose a simple graph-based few-shot learning model that implements a task-driven message passing algorithm. The resulting architecture is trained end-to-end, captures the invariances of the task, such as permutations within the input collections, and offers a good tradeoff between simplicity, generality, performance and sample complexity.

Besides few-shot learning, a related task is the ability to learn from a mixture of labeled and unlabeled examples — semi-supervised learning, as well as *active learning*, in which the learner has the

---

[*]Work done while Victor Garcia was a visiting scholar at New York University

option to request those missing labels that will be most helpful for the prediction task. Our graph-based architecture is naturally extended to these setups with minimal changes in the training design. We validate experimentally the model on few-shot image classification, matching state-of-the-art performance with considerably fewer parameters, and demonstrate applications to semi-supervised and active learning setups.

Our contributions are summarized as follows:

- We cast few-shot learning as a supervised message passing task which is trained end-to-end using graph neural networks.
- We match state-of-the-art performance on Omniglot and Mini-Imagenet tasks with fewer parameters.
- We extend the model in the semi-supervised and active learning regimes.

The rest of the paper is structured as follows. Section 2 describes related work, Sections 3, 4 and 5 present the problem setup, our graph neural network model and the training, and Section 6 reports numerical experiments.

## 2 RELATED WORK

One-shot learning was first introduced by Fei-Fei et al. (2006), they assumed that currently learned classes can help to make predictions on new ones when just one or few labels are available. More recently, Lake et al. (2015) presented a Hierarchical Bayesian model that reached human level error on few-shot learning alphabet recongition tasks.

Since then, great progress has been done in one-shot learning. Koch et al. (2015) presented a deep-learning model based on computing the pair-wise distance between samples using Siamese Networks, then, this learned distance can be used to solve one-shot problems by k-nearest neighbors classification. Vinyals et al. (2016) Presented an end-to-end trainable k-nearest neighbors using the cosine distance, they also introduced a contextual mechanism using an attention LSTM model that takes into account all the samples of the subset $\mathcal{T}$ when computing the pair-wise distance between samples. Snell et al. (2017) extended the work from Vinyals et al. (2016), by using euclidean distance instead of cosine which provided significant improvements, they also build a prototype representation of each class for the few-shot learning scenario. Mehrotra & Dukkipati (2017) trained a deep residual network together with a generative model to approximate the pair-wise distance between samples.

A new line of meta-learners for one-shot learning is rising lately: Ravi & Larochelle (2016) introduced a meta-learning method where an LSTM updates the weights of a classifier for a given episode. Munkhdalai & Yu (2017) also presented a meta-learning architecture that learns meta-level knowledge across tasks, and it changes its inductive bias via fast parametrization. Finn et al. (2017) is using a model agnostic meta-learner based on gradient descent, the goal is to train a classification model such that given a new task, a small amount of gradient steps with few data will be enough to generalize. Lately, Mishra et al. (2017) used Temporal Convolutions which are deep recurrent networks based on dilated convolutions, this method also exploits contextual information from the subset $\mathcal{T}$ providing very good results.

Another related area of research concerns deep learning architectures on graph-structured data. The GNN was first proposed in Gori et al. (2005); Scarselli et al. (2009), as a trainable recurrent message-passing whose fixed points could be adjusted discriminatively. Subsequent works Li et al. (2015); Sukhbaatar et al. (2016) have relaxed the model by untying the recurrent layer weights and proposed several nonlinear updates through gating mechanisms. Graph neural networks are in fact natural generalizations of convolutional networks to non-Euclidean graphs. Bruna et al. (2013); Henaff et al. (2015) proposed to learn smooth spectral multipliers of the graph Laplacian, albeit with high computational cost, and Defferrard et al. (2016); Kipf & Welling (2016) resolved the computational bottleneck by learning polynomials of the graph Laplacian, thus avoiding the computation of eigenvectors and completing the connection with GNNs. In particular, Kipf & Welling (2016) was the first to propose the use of GNNs on semi-supervised classification problems. We refer the reader to Bronstein et al. (2017) for an exhaustive literature review on the topic. GNNs and the analogous Neural Message Passing Models are finding application in many different domains. Battaglia et al.

(2016); Chang et al. (2016) develop graph interaction networks that learn pairwise particle interactions and apply them to discrete particle physical dynamics. Duvenaud et al. (2015); Kearnes et al. (2016) study molecular fingerprints using variants of the GNN architecture, and Gilmer et al. (2017) further develop the model by combining it with set representations Vinyals et al. (2015), showing state-of-the-art results on molecular prediction.

## 3 PROBLEM SET-UP

We describe first the general setup and notations, and then particularize it to the case of few-shot learning, semi-supervised learning and active learning.

We consider input-output pairs $(\mathcal{T}_i, Y_i)_i$ drawn iid from a distribution $P$ of partially-labeled image collections

$$\mathcal{T} = \left\{ \{(x_1, l_1), \dots (x_s, l_s)\}, \{\tilde{x}_1, \dots, \tilde{x}_r\}, \{\bar{x}_1, \dots, \bar{x}_t\} \, ; \, l_i \in \{1, K\}, x_i, \tilde{x}_j, \bar{x}_j \sim \mathcal{P}_l(\mathbb{R}^N) \right\} \, ,$$

and $Y = (y_1, \dots, y_t) \in \{1, K\}^t \, ,$ $\quad$ (1)

for arbitrary values of $s, r, t$ and $K$. Where $s$ is the number of labeled samples, $r$ is the number of unlabeled samples ($r > 0$ for the semi-supervised and active learning scenarios) and $t$ is the number of samples to classify. $K$ is the number of classes. We will focus in the case $t = 1$ where we just classify one sample per task $\mathcal{T}$. $\mathcal{P}_l(\mathbb{R}^N)$ denotes a class-specific image distribution over $\mathbb{R}^N$. In our context, the targets $Y_i$ are associated with image categories of designated images $\bar{x}_1, \dots, \bar{x}_t \in \mathcal{T}_i$ with no observed label. Given a training set $\{(\mathcal{T}_i, Y_i)_i\}_{i \leq L}$, we consider the standard supervised learning objective

$$\min_{\Theta} \frac{1}{L} \sum_{i \leq L} \ell(\Phi(\mathcal{T}_i; \Theta), Y_i) + \mathcal{R}(\Theta) \, ,$$

using the model $\Phi(\mathcal{T}; \Theta) = p(Y \mid \mathcal{T})$ specified in Section 4 and $\mathcal{R}$ is a standard regularization objective.

**Few-Shot Learning** When $r = 0$, $t = 1$ and $s = qK$, there is a single image in the collection with unknown label. If moreover each label appears exactly $q$ times, this setting is referred as the $q$-shot, $K$-way learning.

**Semi-Supervised Learning** When $r > 0$ and $t = 1$, the input collection contains auxiliary images $\tilde{x}_1, \dots, \tilde{x}_r$ that the model can use to improve the prediction accuracy, by leveraging the fact that these samples are drawn from common distributions as those determining the output.

**Active Learning** In the active learning setting, the learner has the ability to request labels from the sub-collection $\{\tilde{x}_1, \dots, \tilde{x}_r\}$. We are interested in studying to what extent this active learning can improve the performance with respect to the previous semi-supervised setup, and match the performance of the one-shot learning setting with $s_0$ known labels when $s + r = s_0$, $s \ll s_0$.

## 4 MODEL

This section presents our approach, based on a simple end-to-end graph neural network architecture. We first explain how the input context is mapped into a graphical representation, then detail the architecture, and next show how this model generalizes a number of previously published few-shot learning architectures.

### 4.1 SET AND GRAPH INPUT REPRESENTATIONS

The input $\mathcal{T}$ contains a collection of images, both labeled and unlabeled. The goal of few-shot learning is to propagate label information from labeled samples towards the unlabeled query image. This propagation of information can be formalized as a posterior inference over a graphical model determined by the input images and labels.

Following several recent works that cast posterior inference using message passing with neural networks defined over graphs Scarselli et al. (2009); Duvenaud et al. (2015); Gilmer et al. (2017), we

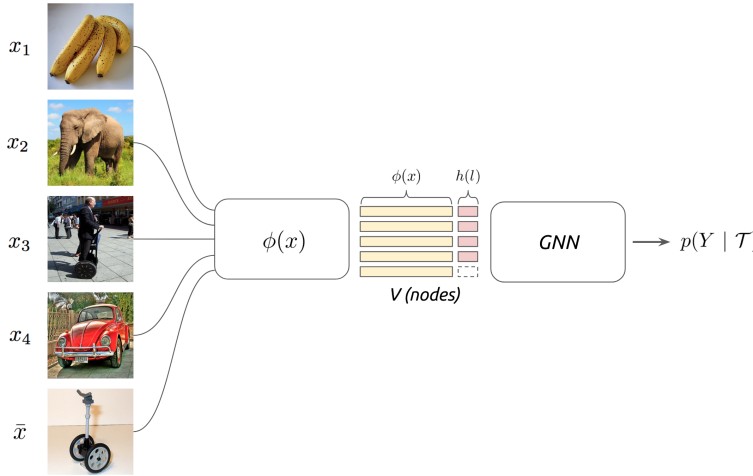

Figure 1: Visual representation of One-Shot Learning setting.

associate $\mathcal{T}$ with a fully-connected graph $G_{\mathcal{T}} = (V, E)$ where nodes $v_a \in V$ correspond to the images present in $\mathcal{T}$ (both labeled and unlabeled). In this context, the setup does not specify a fixed similarity $e_{a,a'}$ between images $x_a$ and $x_{a'}$, suggesting an approach where this similarity measure is learnt in a discriminative fashion with a parametric model similarly as in Gilmer et al. (2017), such as a siamese neural architecture. This framework is closely related to the set representation from Vinyals et al. (2016), but extends the inference mechanism using the graph neural network formalism that we detail next.

## 4.2 GRAPH NEURAL NETWORKS

Graph Neural Networks, introduced in Gori et al. (2005); Scarselli et al. (2009) and further simplified in Li et al. (2015); Duvenaud et al. (2015); Sukhbaatar et al. (2016) are neural networks based on local operators of a graph $G = (V, E)$, offering a powerful balance between expressivity and sample complexity; see Bronstein et al. (2017) for a recent survey on models and applications of deep learning on graphs.

In its simplest incarnation, given an input signal $F \in \mathbb{R}^{V \times d}$ on the vertices of a weighted graph $G$, we consider a family $\mathcal{A}$ of graph intrinsic linear operators that act locally on this signal. The simplest is the *adjacency operator* $A : F \mapsto A(F)$ where $(AF)_i := \sum_{j \sim i} w_{i,j} F_j$ , with $i \sim j$ iff $(i, j) \in E$ and $w_{i,j}$ its associated weight. A GNN layer $\text{Gc}(\cdot)$ receives as input a signal $\mathbf{x}^{(k)} \in \mathbb{R}^{V \times d_k}$ and produces $\mathbf{x}^{(k+1)} \in \mathbb{R}^{V \times d_{k+1}}$ as

$$\mathbf{x}_l^{(k+1)} = \text{Gc}(\mathbf{x}^{(k)}) = \rho \left( \sum_{B \in \mathcal{A}} B \mathbf{x}^{(k)} \theta_{B,l}^{(k)} \right) , \ l = d_1 \ldots d_{k+1} , \tag{2}$$

where $\Theta = \{\theta_1^{(k)}, \ldots, \theta_{|\mathcal{A}|}^{(k)}\}_k$, $\theta_B^{(k)} \in \mathbb{R}^{d_k \times d_{k+1}}$, are trainable parameters and $\rho(\cdot)$ is a point-wise non-linearity, chosen in this work to be a 'leaky' ReLU Xu et al. (2015).

Authors have explored several modeling variants from this basic formulation, by replacing the point-wise nonlinearity with gating operations Duvenaud et al. (2015), or by generalizing the generator family to Laplacian polynomials Defferrard et al. (2016); Kipf & Welling (2016); Bruna et al. (2013), or including $2^J$-th powers of $A$ to $\mathcal{A}$, $A_J = \min(1, A^{2^J})$ to encode $2^J$-hop neighborhoods of each node Bruna & Li (2017). Cascaded operations in the form (2) are able to approximate a wide range of graph inference tasks. In particular, inspired by message-passing algorithms, Kearnes et al. (2016); Gilmer et al. (2017) generalized the GNN to also learn edge features $\tilde{A}^{(k)}$ from the current node hidden representation:

$$\tilde{A}_{i,j}^{(k)} = \varphi_{\tilde{\theta}}(\mathbf{x}_i^{(k)}, \mathbf{x}_j^{(k)}) , \tag{3}$$

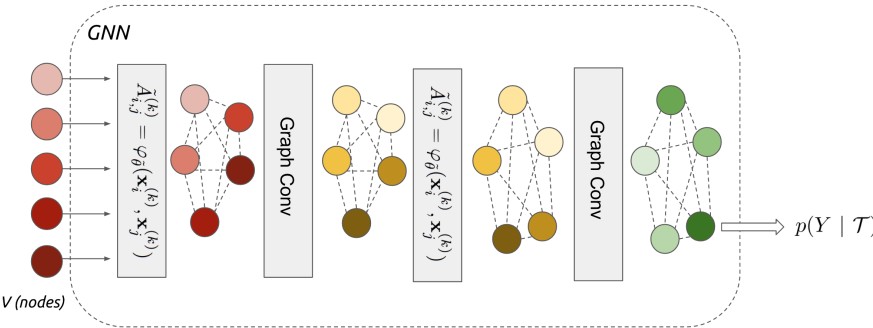

Figure 2: Graph Neural Network ilustration. The Adjacency matrix is computed before every Convolutional Layer.

where $\varphi$ is a symmetric function parametrized with e.g. a neural network. In this work, we consider a Multilayer Perceptron stacked after the absolute difference between two vector nodes. See eq. 4:

$$\varphi_{\tilde{\theta}}(\mathbf{x}_i^{(k)}, \mathbf{x}_j^{(k)}) = \mathrm{MLP}_{\tilde{\theta}}(abs(\mathbf{x}_i^{(k)} - \mathbf{x}_j^{(k)})) \tag{4}$$

Then $\varphi$ is a metric, which is learned by doing a non-linear combination of the absolute difference between the individual features of two nodes. Using this architecture the distance property *Symmetry* $\varphi_{\tilde{\theta}}(a, b) = \varphi_{\tilde{\theta}}(b, a)$ is fulfilled by construction and the distance property *Identity* $\varphi_{\tilde{\theta}}(a, a) = 0$ is easily learned.

The trainable adjacency is then normalized to a stochastic kernel by using a softmax along each row. The resulting update rules for node features are obtained by adding the edge feature kernel $\tilde{A}^{(k)}$ into the generator family $\mathcal{A} = \{\tilde{A}^{(k)}, \mathbf{1}\}$ and applying (2). Adjacency learning is particularly important in applications where the input set is believed to have some geometric structure, but the metric is not known a priori, such as is our case.

In general graphs, the network depth is chosen to be of the order of the graph diameter, so that all nodes obtain information from the entire graph. In our context, however, since the graph is densely connected, the depth is interpreted simply as giving the model more expressive power.

**Construction of Initial Node Features**   The input collection $\mathcal{T}$ is mapped into node features as follows. For images $x_i \in \mathcal{T}$ with known label $l_i$, the one-hot encoding of the label is concatenated with the embedding features of the image at the input of the GNN.

$$\mathbf{x}_i^{(0)} = (\phi(x_i), h(l_i)) , \tag{5}$$

where $\phi$ is a Convolutional neural network and $h(l) \in \mathbb{R}_+^K$ is a one-hot encoding of the label. Architectural details for $\phi$ are detailed in Section 6.1.1 and 6.1.2. For images $\tilde{x}_j, \bar{x}_{j'}$ with unknown label $l_i$, we modify the previous construction to account for full uncertainty about the label variable by replacing $h(l)$ with the uniform distribution over the $K$-simplex: $V_j = (\phi(\tilde{x}_j), K^{-1}\mathbf{1}_K)$, and analogously for $\bar{x}$.

### 4.3 Relationship with Existing Models

The graph neural network formulation of few-shot learning generalizes a number of recent models proposed in the literature.

**Siamese Networks**   Siamese Networks Koch et al. (2015) can be interpreted as a single layer message-passing iteration of our model, and using the same initial node embedding (5) $\mathbf{x}_i^{(0)} = (\phi(x_i), h_i)$ , using a non-trainable edge feature

$$\varphi(\mathbf{x}_i, \mathbf{x}_j) = \|\phi(x_i) - \phi(x_j)\| , \; \tilde{A}^{(0)} = \mathrm{softmax}(-\varphi) ,$$

and resulting label estimation

$$\hat{Y}_* = \sum_j \tilde{A}^{(0)}_{*,j} \langle \mathbf{x}^{(0)}_j, u \rangle \ ,$$

with $u$ selecting the label field from $\mathbf{x}$. In this model, the learning is reduced to learning image embeddings $\phi(x_i)$ whose euclidean metric is consistent with the label similarities.

**Prototypical Networks**   Prototypical networks Snell et al. (2017) evolve Siamese networks by aggregating information within each cluster determined by nodes with the same label. This operation can also be accomplished with a gnn as follows. we consider

$$\tilde{A}^{(0)}_{i,j} = \begin{cases} q^{-1} & \text{if } l_i = l_j \\ 0 & \text{otherwise.} \end{cases}$$

where $q$ is the number of examples per class, and

$$\mathbf{x}^{(1)}_i = \sum_j \tilde{A}^{(0)}_{i,j} \mathbf{x}^{(0)}_j \ ,$$

where $\mathbf{x}^{(0)}$ is defined as in the Siamese Networks. We finally apply the previous kernel $\tilde{A}^{(1)} = \text{softmax}(\varphi)$ applied to $\mathbf{x}^{(1)}$ to yield class prototypes:

$$\hat{Y}_* = \sum_j \tilde{A}^{(1)}_{*,j} \langle \mathbf{x}^{(1)}_j, u \rangle \ .$$

**Matching Networks**   Matching networks Vinyals et al. (2016) use a set representation for the ensemble of images in $\mathcal{T}$, similarly as our proposed graph neural network model, but with two important differences. First, the attention mechanism considered in this set representation is akin to the edge feature learning, with the difference that the mechanism attends always to the same node embeddings, as opposed to our stacked adjacency learning, which is closer to Vaswani et al. (2017). In other words, instead of the attention kernel in (3), matching networks consider attention mechanisms of the form $\tilde{A}^{(k)}_{*,j} = \varphi(\mathbf{x}^{(k)}_*, \mathbf{x}^{(T)}_j)$, where $\mathbf{x}^{(T)}_j$ is the encoding function for the elements of the support set, obtained with bidirectional LSTMs. In that case, the support set encoding is thus computed independently of the target image. Second, the label and image fields are treated separately throughout the model, with a final step that aggregates linearly the labels using a trained kernel. This may prevent the model to leverage complex dependencies between labels and images at intermediate stages.

## 5   TRAINING

We describe next how to train the parameters of the GNN in the different setups we consider: few-shot learning, semi-supervised learning and active learning.

### 5.1   FEW-SHOT AND SEMI-SUPERVISED LEARNING

In this setup, the model is asked only to predict the label $Y$ corresponding to the image to classify $\bar{x} \in \mathcal{T}$, associated with node $*$ in the graph. The final layer of the GNN is thus a softmax mapping the node features to the $K$-simplex. We then consider the Cross-entropy loss evaluated at node $*$:

$$\ell(\Phi(\mathcal{T}; \Theta), Y) = -\sum_k y_k \log P(Y_* = y_k \mid \mathcal{T}) \ .$$

The semi-supervised setting is trained identically — the only difference is that the initial label fields of the node will be filled with the uniform distribution on nodes corresponding to $\tilde{x}_j$.

### 5.2   ACTIVE LEARNING

In the Active Learning setup, the model has the intrinsic ability to query for one of the labels from $\{\tilde{x}_1, \ldots, \tilde{x}_r\}$. The network will learn to ask for the most informative label in order to classify the

sample $\bar{x} \in \mathcal{T}$. The querying is done after the first layer of the GNN by using a Softmax attention over the unlabeled nodes of the graph. For this we apply a function $g(\mathbf{x}_i^{(1)}) \in \mathbb{R}^1$ that maps each unlabeled vector node to a scalar value. Function $g$ is parametrized by a two layers neural network. A Softmax is applied over the $\{1, \ldots, r\}$ scalar values obtained after applying $g$:

$$\text{Attention} = \text{Softmax}(g(\mathbf{x}_{\{1,\ldots,r\}}^{(1)}))$$

In order to query only one sample, we set all elements from the $Attention \in \mathbb{R}^r$ vector to 0 except for one. At test time we keep the maximum value, at train time we randomly sample one value based on its multinomial probability. Then we multiply this sampled attention by the label vectors:

$$w \cdot h(l_{i^*}) = \langle \text{Attention}', h(l_{\{1,\ldots,r\}}) \rangle$$

The label of the queried vector $h(l_{i^*})$ is obtained, scaled by the weight $w \in (0, 1)$. This value is then summed to the current representation $\mathbf{x}_{i^*}^{(1)}$, since we are using dense connections in our GNN model we can sum this $w \cdot h(l_{i^*})$ value directly to where the uniform label distribution was concatenated

$$\mathbf{x}_{i^*}^{(1)} = [\text{Gc}(\mathbf{x}_{i^*}^{(0)}), \mathbf{x}_{i^*}^{(0)}] = [\text{Gc}(\mathbf{x}_{i^*}^{(0)}), (\phi(x_{i^*}), h(l_{i^*}))]$$

After the label has been summed to the current node, the information is forward propagated. This attention part is trained end-to-end with the rest of the network by backpropagating the loss from the output of the GNN.

# 6  EXPERIMENTS

For the few-shot, semi-supervised and active learning experiments we used the Omniglot dataset presented by Lake et al. (2015) and Mini-Imagenet dataset introduced by Vinyals et al. (2016) which is a small version of ILSVRC-12 Krizhevsky et al. (2012). All experiments are based on the $q$-shot, $K$-way setting. For all experiments we used the same values $q$-shot and $K$-way for both training and testing.

Code available at: `https://github.com/vgsatorras/few-shot-gnn`

## 6.1  DATASETS AND IMPLEMENTATION

### 6.1.1  OMNIGLOT

**Dataset:**  Omniglot is a dataset of 1623 characters from 50 different alphabets, each character/class has been drawn by 20 different people. Following Vinyals et al. (2016) implementation we split the dataset into 1200 classes for training and the remaining 423 for testing. We augmented the dataset by multiples of 90 degrees as proposed by Santoro et al. (2016).

**Architectures:**  Inspired by the embedding architecture from Vinyals et al. (2016), following Mishra et al. (2017), a CNN was used as an embedding $\phi$ function consisting of four stacked blocks of {3×3-convolutional layer with 64 filters, batch-normalization, 2×2 max-pooling, leaky-relu} the output is passed through a fully connected layer resulting in a 64-dimensional embedding. For the GNN we used 3 blocks each of them composed by 1) a module that computes the adjacency matrix and 2) a graph convolutional layer. A more detailed description of each block can be found at Figure 3.

### 6.1.2  MINI-IMAGENET

**Dataset:**  Mini-Imagenet is a more challenging dataset for one-shot learning proposed by Vinyals et al. (2016) derived from the original ILSVRC-12 dataset Krizhevsky et al. (2012). It consists of 84×84 RGB images from 100 different classes with 600 samples per class. It was created with the purpose of increasing the complexity for one-shot tasks while keeping the simplicity of a light size dataset, that makes it suitable for fast prototyping. We used the splits proposed by Ravi & Larochelle (2016) of 64 classes for training, 16 for validation and 20 for testing. Using 64 classes for training, and the 16 validation classes only for early stopping and parameter tuning.

**Architecture:**   The embedding architecture used for Mini-Imagenet is formed by 4 convolutional layers followed by a fully-connected layer resulting in a 128 dimensional embedding. This light architecture is useful for fast prototyping:

$1\times\{3\times3$-conv. layer (64 filters), batch normalization, max pool$(2,2)$, leaky relu$\}$,
$1\times\{3\times3$-conv. layer (96 filters), batch normalization, max pool$(2,2)$, leaky relu$\}$,
$1\times\{3\times3$-conv. layer (128 filters), batch normalization, max pool$(2,2)$, leaky relu, dropout$(0.5)\}$,
$1\times\{3\times3$-conv. layer (256 filters), batch normalization, max pool$(2,2)$, leaky relu, dropout$(0.5)\}$,
$1\times\{$ fc-layer (128 filters), batch normalization$\}$.

The two dropout layers are useful to avoid overfitting the GNN in Mini-Imagenet dataset. The GNN architecture is similar than for Omniglot, it is formed by 3 blocks, each block is described at Figure 3.

## 6.2   FEW-SHOT

Few-shot learning experiments for Omniglot and Mini-Imagenet are presented at Table 1 and Table 2 respectively.

We evaluate our model by performing different q-shot, K-way experiments on both datasets. For every few-shot task $\mathcal{T}$, we sample $K$ random classes from the dataset, and from each class we sample $q$ random samples. An extra sample to classify is chosen from one of that $K$ classes.

*Omniglot*: The GNN method is providing competitive results while still remaining simpler than other methods. State of the art results are reached in the 5-Way and 20-way 1-shot experiments. In the 20-Way 1-shot setting the GNN is providing slightly better results than Munkhdalai & Yu (2017) while still being a more simple approach. The TCML approach from Mishra et al. (2017) is in the same confidence interval for 3 out of 4 experiments, but it is slightly better for the 20-Way 5-shot, although the number of parameters is reduced from $\sim$5M (TCML) to $\sim$300K (3 layers GNN).

At *Mini-Imagenet* table we are also presenting a baseline "*Our metric learning + KNN*" where no information has been aggregated among nodes, it is a K-nearest neighbors applied on top of the pair-wise learnable metric $\varphi_\theta(\mathbf{x}_i^{(0)}, \mathbf{x}_j^{(0)})$ and trained end-to-end, this learnable metric is competitive by itself compared to other state of the art methods. Even so, a significant improvement (from 64.02% to 66.41%) can be seen for the 5-shot 5-Way Mini-Imagenet setting when aggregating information among nodes by using the full GNN architecture. A variety of embedding functions $\phi$ are used among the different papers for Mini-Imagenet experiments, in our case we are using a simple network of 4 conv. layers followed by a fully connected layer (Section 6.1.2) which served us to compare between *Our GNN* and *Our metric learning + KNN* and it is useful for fast prototyping. More complex embeddings have proven to produce better results, at Mishra et al. (2017) a deep residual network is used as embedding network $\phi$ increasing the accuracy considerably. Regarding the TCML architecture in Mini-Imagenet, the number of parameters is reduced from $\sim$11M (TCML) to $\sim$400K (3 layers GNN).

| | 5-Way | | 20-Way | |
|---|---|---|---|---|
| **Model** | **1-shot** | **5-shot** | **1-shot** | **5-shot** |
| **Pixels** Vinyals et al. (2016) | 41.7% | 63.2% | 26.7% | 42.6% |
| **Siamese Net** Koch et al. (2015) | 97.3% | 98.4% | 88.2% | 97.0% |
| **Matching Networks** Vinyals et al. (2016) | 98.1% | 98.9% | 93.8% | 98.5% |
| **N. Statistician** Edwards & Storkey (2016) | 98.1% | 99.5% | 93.2% | 98.1% |
| **Res. Pair-Wise** Mehrotra & Dukkipati (2017) | - | - | 94.8% | - |
| **Prototypical Networks** Snell et al. (2017) | 97.4% | 99.3% | 95.4% | 98.8% |
| **ConvNet with Memory** Kaiser et al. (2017) | 98.4% | 99.6% | 95.0% | 98.6% |
| **Agnostic Meta-learner** Finn et al. (2017) | 98.7 ±0.4% | 99.9 ±0.3% | 95.8 ±0.3% | 98.9 ±0.2% |
| **Meta Networks** Munkhdalai & Yu (2017) | 98.9% | - | 97.0% | - |
| **TCML** Mishra et al. (2017) | 98.96% ±0.20% | 99.75% ±0.11% | 97.64% ±0.30% | 99.36% ±0.18% |
| **Our GNN** | 99.2% | 99.7% | 97.4% | 99.0% |

Table 1: Few-Shot Learning — Omniglot accuracies. Siamese Net results are extracted from Vinyals et al. (2016) reimplementation.

| Model | 5-Way | |
| --- | --- | --- |
| | **1-shot** | **5-shot** |
| **Matching Networks** Vinyals et al. (2016) | 43.6% | 55.3% |
| **Prototypical Networks** Snell et al. (2017) | 46.61% ±0.78% | 65.77% ±0.70% |
| **Model Agnostic Meta-learner** Finn et al. (2017) | 48.70% ±1.84% | 63.1% ±0.92% |
| **Meta Networks** Munkhdalai & Yu (2017) | 49.21% ±0.96 | - |
| **Ravi & Larochelle** Ravi & Larochelle (2016) | 43.4% ±0.77% | 60.2% ±0.71% |
| **TCML** Mishra et al. (2017) | 55.71% ±0.99% | 68.88% ±0.92% |
| **Our metric learning + KNN** | 49.44% ±0.28% | 64.02% ±0.51% |
| **Our GNN** | 50.33% ±0.36% | 66.41% ±0.63% |

Table 2: Few-shot learning — Mini-Imagenet average accuracies with 95% confidence intervals.

## 6.3 SEMI-SUPERVISED

Semi-supervised experiments are performed on the 5-way 5-shot setting. Different results are presented when 20% and 40% of the samples are labeled. The labeled samples are balanced among classes in all experiments, in other words, all the classes have the same amount of labeled and unlabeled samples.

Two strategies can be seen at Tables 3 and 4. *"GNN - Trained only with labeled"* is equivalent to the supervised few-shot setting, for example, in the 5-Way 5-shot 20%-labeled setting, this method is equivalent to the 5-way 1-shot learning setting since it is ignoring the unlabeled samples. *"GNN - Semi supervised"* is the actual semi-supervised method, for example, in the 5-Way 5-shot 20%-labeled setting, the GNN receives as input 1 labeled sample per class and 4 unlabeled samples per class.

Omniglot results are presented at Table 3, for this scenario we observe that the accuracy improvement is similar when adding images than when adding labels. The GNN is able to extract information from the input distribution of unlabeled samples such that only using 20% of the labels in a 5-shot semi-supervised environment we get same results as in the 40% supervised setting.

In Mini-Imagenet experiments, Table 4, we also notice an improvement when using semi-supervised data although it is not as significant as in Omniglot. The distribution of Mini-Imagenet images is more complex than for Omniglot. In spite of it, the GNN manages to improve by ∼2% in the 20% and 40% settings.

| Model | 5-Way 5-shot | | |
| --- | --- | --- | --- |
| | **20%-labeled** | **40%-labeled** | **100%-labeled** |
| **GNN - Trained only with labeled** | 99.18% | 99.59% | 99.71% |
| **GNN - Semi supervised** | 99.59% | 99.63% | 99.71% |

Table 3: Semi-Supervised Learning — Omniglot accuracies.

| Model | 5-Way 5-shot | | |
| --- | --- | --- | --- |
| | **20%-labeled** | **40%-labeled** | **100%-labeled** |
| **GNN - Trained only with labeled** | 50.33% ±0.36% | 56.91% ±0.42% | 66.41% ±0.63% |
| **GNN - Semi supervised** | 52.45% ±0.88% | 58.76% ±0.86% | 66.41% ±0.63% |

Table 4: Semi-Supervised Learning — Mini-Imagenet average accuracies with 95% confidence intervals.

## 6.4 ACTIVE LEARNING

We performed Active Learning experiments on the 5-Way 5-shot set-up when 20% of the samples are labeled. In this scenario our network will query for the label of one sample from the unlabeled ones. The results are compared with the *Random* baseline where the network chooses a random sample to be labeled instead of one that maximally reduces the loss of the classification task $\mathcal{T}$.

Results are shown at Table 5. The results of the GNN-Random criterion are close to the Semi-supervised results for 20%-labeled samples from Tables 3 and 4. It means that selecting one random label practically does not improve the accuracy at all. When using the GNN-AL learned criterion, we notice an improvement of $\sim 3.4\%$ for Mini-Imagenet, it means that the GNN manages to correctly choose a more informative sample than a random one. In Omniglot the improvement is smaller since the accuracy is almost saturated and the improving margin is less.

| Method | 5-Way 5-shot 20%-labeled | Method | 5-Way 5-shot 20%-labeled |
|---|---|---|---|
| **GNN - AL** | 99.62% | **GNN - AL** | 55.99% $\pm$1.35% |
| **GNN - Random** | 99.59% | **GNN - Random** | 52.56% $\pm$1.18% |

Table 5: Omniglot (left) and Mini-Imagneet (right), average accuracies are shown at both tables, the GNN-AL is the learned criterion that performs Active Learning by selecting the sample that will maximally reduce the loss of the current classification. The GNN - Random is also selecting one sample, but in this case a random one. Mini-Imagenet results are presented with 95% confidence intervals.

## 7 CONCLUSIONS

This paper explored graph neural representations for few-shot, semi-supervised and active learning. From the meta-learning perspective, these tasks become supervised learning problems where the input is given by a collection or set of elements, whose relational structure can be leveraged with neural message passing models. In particular, stacked node and edge features generalize the contextual similarity learning underpinning previous few-shot learning models.

The graph formulation is helpful to unify several training setups (few-shot, active, semi-supervised) under the same framework, a necessary step towards the goal of having a single learner which is able to operate simultaneously in different regimes (stream of labels with few examples per class, or stream of examples with few labels). This general goal requires scaling up graph models to millions of nodes, motivating graph hierarchical and coarsening approaches Defferrard et al. (2016).

Another future direction is to generalize the scope of Active Learning, to include e.g. the ability to ask questions Rothe et al. (2017), or in reinforcement learning setups, where few-shot learning is critical to adapt to non-stationary environments.

### ACKNOWLEDGMENTS

This work was partly supported by Samsung Electronics (Improving Deep Learning using Latent Structure).

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

## APPENDIX

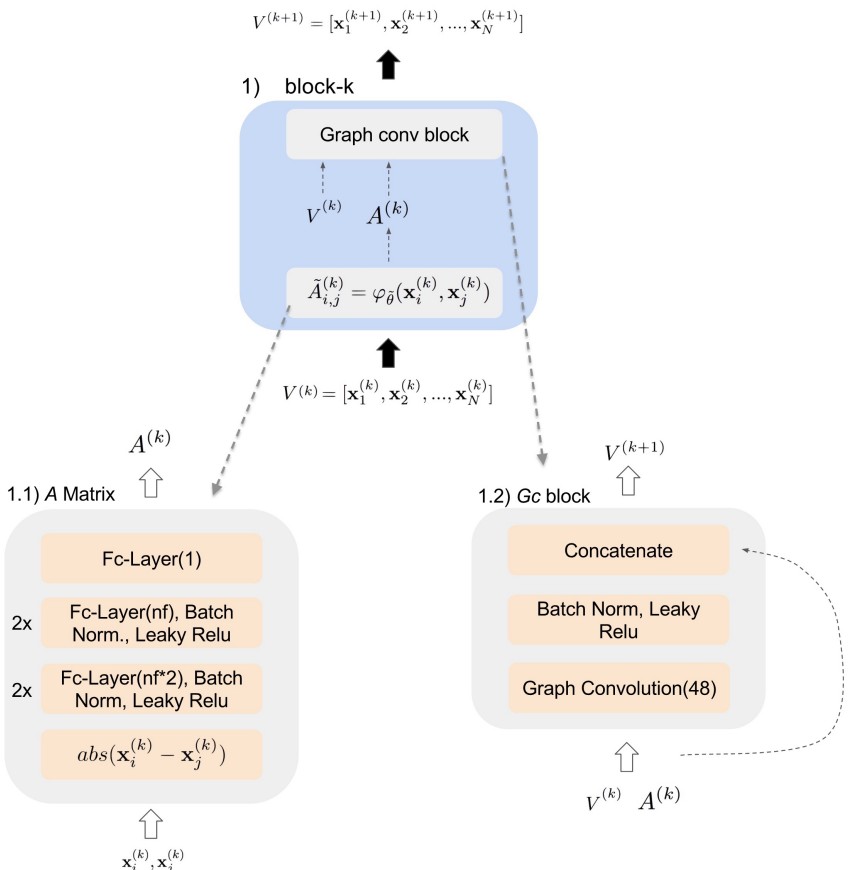

Figure 3: GNN model. Three blue blocks are used for Omniglot and Mini-Imagenet. (nf=96).

