# OpenReview forum: "Few-Shot Learning with Graph Neural Networks"
_ICLR.cc/2018/Conference — Accept (Poster)_

### Official Review · AnonReviewer1 · 2017-11-27
**Good paper with interesting approach**

**Rating:** 7
**Confidence:** 4

**Review:**

This paper studies the problem of one-shot and few-shot learning using the Graph Neural Network (GNN) architecture that has been proposed and simplified by several authors. The data points form the nodes of the graph with the edge weights being learned, using ideas similar to message passing algorithms similar to Kearnes et al and Gilmer et al. This method generalizes several existing approaches for few-shot learning including Siamese networks, Prototypical networks and Matching networks. The authors also conduct experiments on the Omniglot and mini-Imagenet data sets, improving on the state of the art.

There are a few typos and the presentation of the paper could be improved and polished more. I would also encourage the authors to compare their work to other unrelated approaches such as Attentive Recurrent Comparators of Shyam et al, and the Learning to Remember Rare Events approach of Kaiser et al, both of which achieve comparable performance on Omniglot. I would also be interested in seeing whether the approach of the authors can be used to improve real world translation tasks such as GNMT.

---

> ### Author Response · Authors · 2017-12-22
> **Answer**
>
>
> Thank you for the review,
>
>
> > I would also encourage the authors to compare their work to other unrelated approaches such as Attentive Recurrent Comparators of Shyam et al, and the Learning to Remember Rare Events.
>
> We added the results from "Learning to Remember Rare Events" to our table. Regarding "Attentive Recurrent Comparators" we didn't add it due to some differences in the data augmentation method (translation, mirroring and shearing) and the use of Wide Resnets as feature extractors.
>
>
> > I would also be interested in seeing whether the approach of the authors can be used to improve real world translation tasks such as GNMT.
>
> I guess the Graph Neural Network could replace the attentive module for Neural Machine Translation systems that search for parts over the source sentence. Maybe it could exploit more complex relationships among the words of the input sentence. I don't know if it has been already tried.
>
>
> Next, I list some of the main modifications that we have done to the paper:
>
> - We updated Omniglot results using the same data augmentation protocol than other papers and results are now more competitive with the state of the art.
>
> - We considerably improved Mini-Imagenet results by regularizing better (using dropout and early stopping.)
>
> - We improved the writing, figures and we corrected some typos.

---

> > ### Public Comment · ~Pranav_Shyam1 · 2017-12-30
> > **Clarification: ARCs don't use 105x105 images**
> >
> > >  Regarding the Attentive Recurrent Comparators, we couldn't add it because they are using 105x105 resolution version of the images, which makes it hard to be compared with 28x28 resolution versions.
> >
> > The results in the paper are using 32x32 images. Results in Koch et al uses 105x105 images and that is highlighted in our work.

---

> > > ### Author Response · Authors · 2017-12-30
> > > **Answer to Clarification**
> > >
> > >
> > > Hi Parnav, thank you for the clarification.
> > >
> > > I thought 32x32 was the resolution of the attentive patch instead of the image size, but that was wrong, sorry for that. I edited my first comment in order to avoid confusing future readers. Even though, we didn't add the paper due to other differences in the data augmentation and the embedding network now commented in the first answer.

---

### Official Review · AnonReviewer3 · 2017-11-28
**Novel idea for few-shot learning**

**Rating:** 7
**Confidence:** 4

**Review:**

This paper introduces a graph neural net approach to few-shot learning. Input examples form the nodes of the graph and edge weights are computed as a nonlinear function of the absolute difference between node features. In addition to standard supervised few-shot classification, both semi-supervised and active learning task variants are introduced. The proposed approach captures several popular few-shot learning approaches as special cases. Experiments are conducted on both Omniglot and miniImagenet datasets.

Strengths
- Use of graph neural nets for few-shot learning is novel.
- Introduces novel semi-supervised and active learning variants of few-shot classification.

Weaknesses
- Improvement in accuracy is small relative to previous work.
- Writing seems to be rushed.

The originality of applying graph neural networks to the problem of few-shot learning and proposing semi-supervised and active learning variants of the task are the primary strengths of this paper. Graph neural nets seem to be a more natural way of representing sets of items, as opposed to previous approaches that rely on a random ordering of the labeled set, such as the FCE variant of Matching Networks or TCML. Others will likely leverage graph neural net ideas to further tackle few-shot learning problems in the future, and this paper represents a first step in that direction.

Regarding the graph, I am wondering if the authors can comment on what scenarios is the graph structure expected to help? In the case of 1-shot, the graph can only propagate information about other classes, which seems to not be very useful.

Though novel, the motivation behind the semi-supervised and active learning setup could use some elaboration. By including unlabeled examples in an episode, it is already known that they belong to one of the K classes. How realistic is this set-up and in what application is it expected that this will show up?

For active learning, the proposed method seems to be specific to the case of obtaining a single label. How can the proposed method be scaled to handle multiple requested labels?

Overall the paper is well-structured and related work covers the relevant papers, but the details of the paper seem hastily written.

In the problem set-up section, it is not immediately clear what the distinction between s, r, and t is. Stating more explicitly that s is for the labeled data, etc. would make this section easier to follow. In addition, I would suggest stating the reason why t=1 is a necessary assumption for the proposed model in the few-shot and semi-supervised cases.

Regarding the Omniglot dataset, Vinyals et al. (2016) augmented the classes so that 4,800 classes were used for training and 1,692 for test. Was the same procedure done for the experiments in the paper? If yes, please update 6.1.1 to make this distinction more clear. If not, please update the experiments to be consistent with the baselines.

In the experiments, does the \varphi MLP explicitly enforce symmetry and identity or is it learned?

Regarding the Omniglot baselines, it appears that Koch et al. (2015), Edwards & Storkey (2016), and Finn et al. (2017) use non-standard class splits relative to the other methods. This should probably be noted.

The results for Prototypical Networks appear to be incorrect in the Omniglot and Mini-Imagenet tables. According to Snell et al. (2017) they should be 49.4% and 68.2% for miniImagenet. Moreover, Snell et al. (2017) only used 64 classes for training instead of 80 as utilized in the proposed approach. Given this, I am wondering if the authors can comment on the performance difference in the 5-shot case, even though Prototypical Networks is a special case of GNNs?

For semi-supervised and active-learning results, please include error bars for the miniImagenet results. Also, it would be interesting to see 20-way results for Omniglot as the gap between the proposed method and the baseline would potentially be wider.

Other Comments:

- In Section 4.2, Gc(.) is defined in Equation 2 but not mentioned in the text.
- In Section 4.3, adding an equation to clarify the relationship with Matching Networks would be helpful.
- I believe there is a typo in section 4.3 in that softmax(\varphi) should be softmax(-\varphi), so that more similar pairs will be more heavily weighted.
- The equation in 5.1 appears to be missing a minus sign.

Overall, the paper is novel and interesting, though the clarity and experimental results could be better explained.

EDIT: I have read the author's response. The writing is improved and my concerns have largely been addressed. I am therefore revising my rating of the paper to a 7.

---

> ### Author Response · Authors · 2017-12-07
> **Answer part 2**
>
>
>
> > In the experiments, does the \varphi MLP explicitly enforce symmetry and identity or is it learned?
>
> The \varphi(a,b) = MLP(abs(a-b)) explicity enforces symmetry due to the absolute value.
> The identity is easily learned since the input to the MLP will always be the same vector (a vector of zeros) when a==b.
> I have rewritten this line to clarify which property is enforced and which one is easily learned.
>
>
> > Regarding the Omniglot baselines, it appears that Koch et al. (2015), Edwards & Storkey (2016), and Finn et al. (2017) use non-standard class splits relative to the other methods. This should probably be noted.
>
> - From Koch et al. (2015) we are using the results from Vinyals et al. (2016) reimplementation that is using the common class splits. I added a note explaining it in the caption of the table.
>
> - I checked again the paper from Finn et al. (2017). Based on what I read, they are using the same configuration splits, 1200 training, 423 testing and augmented by multiples of 90 degrees. I add the paper url at the end of this answer. Same for Edwards & Storkey (2016). Correct me if I am wrong.
>
>
> > The results for Prototypical Networks appear to be incorrect in the Omniglot and Mini-Imagenet tables. According to Snell et al. (2017) they should be 49.4% and 68.2% for miniImagenet. Given this, I am wondering if the authors can comment on the performance difference in the 5-shot case, even though Prototypical Networks is a special case of GNNs?
>
> In order to use the same evaluation procedure across papers, all results are evaluated using the same K-way q-shot conditions for both training an test, in other words, a network that for example evaluates a 20-way 1-shot experiment has been trained on 20-way 1-shot tasks. This was the evaluation procedure presented by (Vinyals et al.) and followed by later works. In Prototypical Networks these results are reported in the Appendix. (Mishra et al.) is also reporting these results from (Snell et al) in their comparison. We chose to use the same evaluation procedure across papers.
>
>
> > Moreover, Snell et al. (2017) only used 64 classes for training instead of 80 as utilized in the proposed approach.
>
> We modified it, in the results of the last update, the network is trained with the 64 training classes. The 16 validation classes are only used for early stopping and parameter tuning.
>
>
> > For semi-supervised and active-learning results, please include error bars for the miniImagenet results. Also, it would be interesting to see 20-way results for Omniglot as the gap between the proposed method and the baseline would potentially be wider.
>
> We added the error bars.
>
>
> > Other Comments:
> > In Section 4.2, Gc(.) is defined in Equation 2 but not mentioned in the text.
>
> Solved
>
>
> > In Section 4.3, adding an equation to clarify the relationship with Matching Networks would be helpful.
>
> Done
>
>
> > I believe there is a typo in section 4.3 in that softmax(\varphi) should be softmax(-\varphi), so that more similar pairs will be more heavily weighted.
>
> Solved
>
>
> > The equation in 5.1 appears to be missing a minus sign.
>
> Solved
>
>
> We improved Mini-Imagenet results by regularizing better (using dropout and early stopping.)
>
>
> - Finn et al. (2017) https://arxiv.org/pdf/1703.03400.pdf
> - Edwards & Storkey (2016) https://arxiv.org/pdf/1606.02185.pdf

---

> ### Author Response · Authors · 2017-12-07
> **Answer part 1**
>
>
> First of all thank you for the review and comments.
>
>
> > Regarding the graph, I am wondering if the authors can comment on what scenarios is the graph structure expected to help? In the case of 1-shot, the graph can only propagate information about other classes, which seems to not be very useful.
>
> I would like to point out two main strengths of the GNN method that we are proposing:
> 1) A different metric from the euclidean is learned at every layer.
> 2) GNN can handle contextual information by aggregating information from the neighbor nodes based on the weights of the adjacency matrix learned in (1).
>
> Based on our experiments, propagating information from other classes seems more useful when the number of classes is larger than one, specifically we notice the largest improvement in Mini-Imagenet 5-way 5-shot. But the metric that is learned at every layer also provides strong results for the case of one-shot, specially we notice it in the Omniglot dataset in 5-way 1-shot and 20-way 1-shot. The Graph structure also allows us to run in the semi-supervised and active learning scenarios.
>
>
> > Though novel, the motivation behind the semi-supervised and active learning setup could use some elaboration. By including unlabeled examples in an episode, it is already known that they belong to one of the K classes. How realistic is this set-up and in what application is it expected that this will show up?
>
> One example application that comes to my mind for the semi-supervised scenario would be building a face recognition system from Facebook profiles. A user may have uploaded dozens of images with other people and maybe just some of the pictures are labeled. It could be possible to use all the faces from the pictures that the user uploaded even if they are not labeled together with the few labeled ones to build a few-shot classifier for that user.
>
> I hope the open community will be able to find new and better applications.
>
>
> > For active learning, the proposed method seems to be specific to the case of obtaining a single label. How can the proposed method be scaled to handle multiple requested labels?
>
> In the proposed method, we are uncovering one of the labels choosing from a softmax distribution in a particular layer. It would be possible to uncover multiple labels by choosing more than one, it woud also be possible to uncover multiple labels at multiple layers. Our main aim here was to test out how feasible is to learn to do Active learning from the classification loss using an end to end structure like a GNN instead of using handcrafted methods like Uncertainty Sampling. Scaling it to larger datasets can be hard to optimize, we leave it for the future and we present it as a prove of concept.
>
>
> > In the problem set-up section, it is not immediately clear what the distinction between s, r, and t is.
>
> We have rewritten this section more accurately.
>
>
> > Regarding the Omniglot dataset, Vinyals et al. (2016) augmented the classes so that 4,800 classes were used for training and 1,692 for test. Was the same procedure done for the experiments in the paper? If yes, please update 6.1.1 to make this distinction more clear. If not, please update the experiments to be consistent with the baselines.
>
> Thanks a lot for this point. Our data augmentation implementation was only for the training classes. We updated the paper results implementing the data augmentation also on the test classes as it is done in other works and the results are now better and more competitive with the state of the art.
>
> The answer continues in the following message "Answer part 2"

---

### Official Review · AnonReviewer2 · 2017-11-30

**Rating:** 7
**Confidence:** 4

**Review:**

This paper proposes to use graph neural networks for the purpose of few-shot learning, as well as semi-supervised learning and active learning. The paper first relies on convolutional neural networks to extract image features. Then, these image features are organized in a fully connected graph. Then, this graph is processed with an graph neural network framework that relies on modelling the differences between features maps, \propto \phi(abs(x_i-x_j)).  For few-shot classification then the cross-entropy classification loss is used on the node.

The paper has some interesting contributions and ideas, mainly from the point of view of applications, since the basic components (convnets, graph neural networks) are roughly similar to what is already proposed. However, the novelty is hurt by the lack of clarity with respect to the model design.

First, as explained in 5.1 a fully connected graph is used (although in Fig. 2 the graph nodes do not have connections to all other nodes). If all nodes are connected to all nodes, what is the different of this model from a fully connected, multi-stream networks composed of S^2 branches? To rephrase, what is the benefit of having a graph structure when all nodes are connected with all nodes. Besides, what is the effect when having more and more support images? Is the generalization hurt?

Second, it is not clear whether the label used as input in eq. (4) is a model choice or a model requirement. The reason is that the label already appears in the loss of the nodes  in 5.1. Isn't using the label also as input redundant?

Third, the paper is rather vague or imprecise at points.  In eq. (1) many of the notations remain rather unclear until later in the text (and even then they are not entirely clear). For instance, what is s, r, t.

The experimental section is also ok, although not perfect.  The proposed method appears to have a modest improvement for few-shot learning. However, in the case of  active learning and semi-supervised learning the method is not compared to any baselines (other than the random one), which makes conclusions hard to reach.

In general, I tend to be in favor of accepting the paper if the authors have persuasive answers and provide the clarifications required.

---

> ### Author Response · Authors · 2017-12-07
> **Answer**
>
>
> First of all, thank you for the review and comments.
>
>
> > As explained in 5.1 a fully connected graph is used (although in Fig. 2 the graph nodes do not have connections to all other nodes). If all nodes are connected to all nodes, what is the different of this model from a fully connected, multi-stream networks composed of S^2 branches?
>
> The graph is defined as fully connected, but the connection weights are different among nodes. In other words, the adjacency matrix $A$ is not a binary matrix formed by 0s and 1s, instead of that, every value of the adjacency matrix $A[i,j]$ is a Real number that ranges from 0 to 1. These values are computed by the network at every layer before applying each Graph Convolution.
>
> Our Graph Neural Network (GNN) performs a particular operation on each node of the Graph and another operation is applied on the neighbors of that node that are averaged using the weights of the Adjacency matrix. This behaviour is easier to represent unfolding the equation 2 from the paper:
>
> x^{k+1} = Gc(x^{k}) = ρ(I·x^{k}·\theta_1^{k} + A·x^{k}·\theta_2^{k}).
>
> Where $k$ is the layer index; $x^{k}$ are the nodes structured into a 2-dimensional (Number_nodes by Number_features) matrix; $I$ is the identity matrix; $A$ is the adjacent matrix; and \theta_1^{k} and \theta_2^{k} are the vectors of learnable parameters of dimensionality (Number_features,) for these two operations.
>
> The matrices $I$ and $A$ are called operators, in our paper are represented with the following term  $\mathcal{A} = {A^{k}, I}$
>
> Once it has been clarified, we can notice that the mechanism of a GNN is different from a multi-stream network composed by S^2 branches. In a GNN the same operation is convolved over the nodes of a layer and at every node the information of the neighbors is aggregated based on the Adjacency matrix, the parameters to learn at every layer will only be theta_1 and theta_2. Based on my knowledge about multi-stream networks, the number of parameters to handle S^2 branches would be radically larger, the mechanism to aggregate the information from the other nodes would also be different. I hope this explanation clarified this point.
>
> Regarding Fig. 2 I updated it connecting all the nodes. In the previous version I had removed some of the connections to clearly see how they change at every layer, but it was probably misleading.
>
>
> > To rephrase, what is the benefit of having a graph structure when all nodes are connected with all nodes. Besides, what is the effect when having more and more support images? Is the generalization hurt?
>
> We can distinguish to main benefits of the GNN:
> 1) A different metric from the euclidean is learned at every layer.
> 2) GNN can handle contextual information by aggregating information from the other nodes to the current one based on the weights of the adjacency matrix learned in (1).
>
> About the generalization, the number of parameters of the GNN is independent from the number of support images, therefore, increasing the number of support images will not overparametrize the network avoiding the risk to overfit.
>
>
> > Second, it is not clear whether the label used as input in eq. (4) is a model choice or a model requirement. The reason is that the label already appears in the loss of the nodes in 5.1. Isn't using the label also as input redundant?
>
> In a few-shot task we have: 1) A support subset of labeled images and 2) An unlabeled image to classify. We input (1) the label of the subset of labeled images and we predict (2) the label of the image to classify which appears in the loss. Therefore it is a model requirement for the few-shot scenario. We modified this explanation to be clearer. In the semi-supervised scenario we just input the label for some of the samples, therefore in this case, it is only a requirement for the labeled samples.
>
>
> > Third, the paper is rather vague or imprecise at points.  In eq. (1) many of the notations remain rather unclear until later in the text (and even then they are not entirely clear). For instance, what is s, r, t.
>
> We have rewritten this section in order to be clearer.
>
>
> > The experimental section is also ok, although not perfect.  The proposed method appears to have a modest improvement for few-shot learning. However, in the case of  active learning and semi-supervised learning the method is not compared to any baselines (other than the random one), which makes conclusions hard to reach.
>
> We updated the Omniglot results using the same data augmentation protocol than other papers and results are more competitive with the state of the art now.
> We also improved Mini-Imagenet results by better regularizing the network using dropout and early stopping.

---

### Public Comment · (anonymous) · 2017-11-28
**Experimental Settings**

Dear Authors,

This paper introduced adding a graph neural network on top of a feature extractor to conduct comparisons between inputs.
The idea of adding a module to combine features from multiple inputs have been studied extensively in this field (Snell et al. and Mishra et al.).
Thus, I feel that it is critical for this paper to prove empirical strength of the proposed architectural extension.

With regard to experiments, I have few questions and proposals to make it better.
First, why is the score for the Prototypical Networks much lower than what is reported in the paper?
For instance, 5-way 1-shot score of miniImageNet, the Prototypical Networks is reported to be 49.42% (conf interval 0.78%), but you reported it as 46.61%.
By taking confidence interval into account, there is no statistically significant difference between your method (49.8% with conf interval 0.22%) and theirs.
I find this a crucial problem in the experiment section because your method has generalized from the Prototypical Networks, and you claim that the extra flexibility has led to improvement in performance.
By the way, the score by Snell et al. is reproducible (I have done it myself for 5-way 1-shot setting of MiniImageNet, although I have failed to do so for 5-way 5-shot).

Second, this is only a suggestion, but it would be nice if you add an experiment with ResNet architecture used by TCML (Mishra et al.) for MiniImageNet.
Your experimental results are not comparable to the current state of the art with weaker feature extractor than TCML.

- Snell et al. (2017). "Prototypical Networks for Few-shot Learning." (https://arxiv.org/abs/1703.05175)
- Mishra et al. (2017). "Meta-Learning with Temporal Convolutions." (https://arxiv.org/abs/1707.03141)

---

> ### Public Comment · (anonymous) · 2017-11-28
> **Answer**
>
>
> First of all, thank you very much for the comment and remarks.
>
> > Why is the score for the Prototypical Networks much lower than what is reported in the paper?
>
> In order to use the same evaluation procedure across papers, all results are evaluated using the same K-way q-shot conditions for both training an test, in other words, a network that for example evaluates a 20-way 1-shot experiment has been trained on 20-way 1-shot tasks. This was the evaluation procedure presented by (Vinyals et al.) and followed by later works. In Prototypical Networks these results are reported in the Appendix. (Mishra et al.) is also reporting these results from (Snell et al) in their comparison.
>
>
> > By taking confidence interval into account, there is no statistically significant difference between your method (49.8% with conf interval 0.22%) and theirs. I find this a crucial problem in the experiment section because your method has generalized from the Prototypical Networks, and you claim that the extra flexibility has led to improvement in performance.
>
> It is true that MiniImagenet results are in the same confidence interval than Prototypical Networks, despite this, Graph Neural Networks improve significantly in Omniglot dataset compared to Prototypical Networks. And they are also able to do semi-supervised and active learning which is not possible with the prototypical setting. This is why we claim the extra flexibility. Graph Neural Networks have two main interesting properties in few-shot learning: 1) They learn a different metric from the euclidean at every layer. 2) They can handle contextual information. Roughly speaking, the point (1) seems to be useful for Omniglot and the point (2) seems to be useful for MiniImagenet.
>
> Edit: We've improved the results for MiniImagenet to 50.33% (with conf interval 0.36%) in the 5-way 1-shot setting
>
>
> > Second, this is only a suggestion, but it would be nice if you add an experiment with ResNet architecture used by TCML (Mishra et al.) for MiniImageNet.
>
> We will consider it in order to get a better comparison with Mishra et al. (2017).
>
>
> - Vinyals et al. (2016) "Matching Networks for One Shot Learning." (https://arxiv.org/pdf/1606.04080.pdf)
> - Snell et al. (2017). "Prototypical Networks for Few-shot Learning." (https://arxiv.org/abs/1703.05175)
> - Mishra et al. (2017). "Meta-Learning with Temporal Convolutions." (https://arxiv.org/abs/1707.03141)

---

> > ### Public Comment · (anonymous) · 2017-11-29
> > **Reply to Answer**
> >
> > > In order to use the same evaluation procedure across papers, all results are evaluated using the same K-way q-shot conditions for both training an test, in other words, a network that for example evaluates a 20-way 1-shot experiment has been trained on 20-way 1-shot tasks.
> >
> > I am not really satisfied with the protocol you used.
> > There is no constraint on how you train the network for few-shot image classification,
> > thus it is fine to use settings different in training and testing.
> > I do not think that the fact that Mishra et al. does similar report justifies this problem.
> >
> >
> > > It is true that MiniImagenet results are in the same confidence interval than Prototypical Networks, despite this, Graph Neural Networks improve significantly in Omniglot dataset compared to Prototypical Networks. And they are also able to do semi-supervised and active learning which is not possible with the prototypical setting.
> >
> > Thank you for your answer.
> > Although I had a strong concern about the supervised training result for MiniImageNet, your comments led me to conclude that there are strength in your proposal.

---

> > > ### Public Comment · (anonymous) · 2017-11-29
> > > **Answer 2**
> > >
> > >
> > > > I am not really satisfied with the protocol you used. There is no constraint on how you train the network for few-shot image classification, thus it is fine to use settings different in training and testing.
> > >
> > > I agree with you there is no constraint in how you train the network, but we chose to use the same evaluation protocol than other papers in order to do a fairer evaluation.

---

### Public Comment · (anonymous) · 2017-12-14
**Some insight on the paper**

Greetings, could you give me some insight on the following points:
Equation (2):
* What do d_k and d_k+1 stands for? If they are the point dimension for the k-stage and k+1-stage how can you iterate over the [l] index (which goes from 1 to d_k+1) for both?

Figure (8):
* Why you use a concatenation? is it suppose to represent the 1 in the generator family? Shouldn't it be added instead of concatenated?

Thanks in advance

---

> ### Author Response · Authors · 2017-12-15
> **Answer**
>
>
> Thanks for reading the paper,
>
>
> > What do d_k and d_k+1 stands for?
>
> As you said, they are the point dimension for the k-stage and k+1-stage.
>
>
> > If they are the point dimension for the k-stage and k+1-stage how can you iterate over the [l] index (which goes from 1 to d_k+1) for both?
>
> You can see that x^{(k)}$ and $\theta^{(k)}$ are also indexed by $k$, therefore, this $k$ is also changing the value of [l] which also depends on $k$. In practice we just mean that at every layer the input dimensionality is d_k, and the output is d_k+1, same as in CNNs.
>
>
> > Why you use a concatenation? is it suppose to represent the 1 in the generator family? Shouldn't it be added instead of concatenated?
>
> The concatenation is independent from the Graph operation, we are just concatenating the input layer to the new outputs, this is applied to other types of Neural Networks. It was introduced by "Densely Connected Convolutional Networks", https://arxiv.org/abs/1608.06993

---

> > ### Author Response · Authors · 2017-12-15
> > **typo in the answer**
> >
> > Thanks again for your comment.
> >
> > We apologize, there is a typo in the equation (and in the previous answer). The subscript "l" is indexing over output feature maps, not over input feature maps. We will fix this in the document.
> >
> > Best,
> >
> > Authors

---

### Decision · Program_Chairs · 2018-01-29
**ICLR 2018 Conference Acceptance Decision**

**Decision:**

Accept (Poster)

**Comment:**

All reviewers agree that the proposed method is novel and experiments do a good job in establishing its value for few-shot learning. Most the concerns raised by the reviewers on experimental protocols have been addressed in the author response and revised version.